# Nonlinear Modulation of Plasmonic Resonances in Graphene-Integrated Triangular Dimers at Terahertz Frequencies

**DOI:** 10.3390/ma12152466

**Published:** 2019-08-02

**Authors:** Quan Li, Shuang Wang, Tai Chen

**Affiliations:** 1School of Electronic Engineering, Tianjin University of Technology and Education, Tianjin 300222, China; 2National-Local Joint Engineering Laboratory of Intelligent Manufacturing Oriented Automobile Die & Mould, Tianjin University of Technology and Education, Tianjin 300222, China

**Keywords:** metamaterials, graphene, terahertz

## Abstract

Metamaterials made from artificial subwavelength structures hold great potential in designing functional devices at microwave, terahertz, infrared, and optical frequencies. In this work, we study the active switching effect of the plasmonic resonance modes in triangular dimer (DTD) structure using graphene in the terahertz regime. The sole DTD structure can only support a dipolar bonding dimer plasmonic (BDP) mode, whose field enhancement factor at the gap center can reach 67.4. However, with a metallic junction in the dimer, the BDP mode switches to a charge transfer plasmonic (CTP) mode. When changing the metallic junction to a graphene stripe, an active modulation effect of the CTP mode can be realized by altering the nonlinear conductivity of graphene through strong-field terahertz incidence. The proposed design is quite promising in terahertz sensing, amplitude switching and nonlinear effect enhancement, etc.

## 1. Introduction

Metamaterials, which possess unique electromagnetic properties far beyond those achieved by naturally formed materials, have drawn huge attention in recent years. The functions of metamaterials can be designed nearly at will by properly engineering the subwavelength structures and their arrangement at frequencies of interest. In the terahertz regime where the current mature technologies are almost not applicable, metamaterials serve as a potential tool in realizing functional devices. Many metamaterial devices have been well demonstrated, including filters [1,2,3], sensors [4,5], wavefront modulators [6,7,8], and nonlinear devices [9,10,11]. It is clear to see that those functions were all achieved based on utilizing certain resonances, such as LC resonance, Fano resonance, dipolar resonance, etc. However, those devices mainly functioned in a passive manner. Once the structures were determined, the performances were fixed. New ways of realizing active metamaterials devices are highly demanded for future compact and convenient applications.

Recently, active terahertz metamaterial devices have become a hot topic. By integrating functional materials whose conductivity can be externally changed, such as silicon on sapphire, vanadium dioxide, and gallium arsenide, etc., into the structural unit cells, the responses could be actively tuned by optical pump, heating, and bias voltage, respectively [12,13,14,15,16]. Beside those materials, graphene, as a two-dimensional material with fast tunable optical conductivity affected by either a optical pump or bias voltage, has become a new favorite material for researchers. To date, many graphene based active metamaterial devices have been demonstrated in the terahertz regime, including terahertz diodes, tunable slow-light devices, chiral switches, and so on [17,18,19,20,21,22,23,24,25,26].

In this paper, we study the mode switching effect by employing planar metamaterials composed of arrays of metallic double triangular dimers (DTD). We find that when the dimer is collected by metal or dynamic material junction with high conductivity, the supported resonance mode would transit from the BDP mode to the CTP mode. By changing the junction with graphene stripe, dynamic mode transition effect can be observed. We also observed that the DTD structure could enhance the field at the gap, which becomes stronger when decreasing the gap distance. It is supposed that such an effect could be utilized to realize active mode switching by tuning the graphene conductivity using nonlinear effect of graphene under strong-field terahertz incidence. The presented results offer a possible avenue to realize highly sensitive sensors, nonlinear devices, and active amplitude modulators in the terahertz regime.

## 2. Sample Design and Characterization

The basic unit cell of the metamaterial is illustrated in Figure 1a, which is a DTD structure consisting of two equilateral triangles with a period of *P* = 100 μm. The corresponding geometric parameters of the DTD are illustrated in Figure 1b, in which the length *L* = 36 μm and the gap distance *g* = 6 μm. The structures are made from 200 nm-thickness aluminum on a *p*-type silicon wafer with a thickness of *t* = 640 μm. The samples were fabricated by employing conventional lithography. Each sample has the size of 1 cm × 1 cm, which contains 10^4^ unit cells.

The samples were experimentally characterized using a broadband terahertz time-domain spectroscopy (THz-TDS) [27]. During the whole measurement, the humidity was kept less than 5% to eliminate water absorption. The amplitude transmission was defined by $\left|\tilde{t}(\omega )\right|=\left|{\tilde{E}}_{s}(\omega )/{\tilde{E}}_{r}(\omega )\right|$, where ${\tilde{E}}_{s}(\omega )$ and ${\tilde{E}}_{r}(\omega )$ are the Fourier transformed spectra of the time-domain signals passing through the sample and the reference (a bare silicon substrate), respectively.

## 3. Results and Discussion

The red curve in Figure 2a illustrate the measured amplitude transmission spectra of the DTD sample under *x*-polarized incidence. It can be seen that there is a broad resonance at around 1.71 THz. However, when the DTD structures were connected with a 10 μm × 6 μm aluminum junction (denote them as DTDJ structures), as indicated by the inset of Figure 2c, the transmission under *x*-polarized incidence became completely different. As shown in the red curve in Figure 2c, a sharp and strong resonance at 0.58 THz emerges in the measured result whose quality (*Q*) factor is 5.26, while the original resonance mode at 1.71 THz disappears. We also studied the cases under *y*-polarized incidences, as illustrated in the blue curves in Figure 2a,c. Such a mode transition effect was not observed. Both the DTD and DTDJ structures exhibited a broad resonance mode at around 1.75 THz.

In order to gain in-depth understanding of the resonance modes of the DTD and DTDJ structures, finite-element time-domain (FDTD) method was carried out to simulate the transmission spectra. In the simulation, the metal was modeled as a material with a conductivity of 3.72 × 10^7^ S/m, and the substrate was modeled as loss free silicon with relative permittivity of 11.9. Periodic boundary conditions were applied at both x and y directions. A plane wave (*x*-polarized or *y*-polarized) was set to normally illuminate onto the structure. The corresponding simulated transmission spectra of the DTD and DTDJ structures are respectively illustrated in Figure 2b,d, respectively. It can be seen clearly that the simulated results under both *x*- and *y*-polarized incidences are in good agreement with the experimental results in Figure 2a,c, which reveals the validity of the applied simulation method. 

To investigate the resonant behaviors of the observed resonance modes, we also simulated the electric field distributions at the corresponding resonance frequencies, as illustrated by the simulated electric field distributions at the structure plane in the insets of Figure 2b,d. For the DTD structure, the field maxima have opposite signs at the ends of the gaps under *x*-polarized incidence. Thus, it can be seen as a BDP mode resonance [28,29,30,31]. In this mode, the double triangles works as two in-phase dipoles along the *x* direction. A large amount of carriers with opposite sign can be accumulated at the ends of the gap, resulting in a strong field enhancement at the gap. For the DTDJ structure, the field maxima are at the side ends of the triangles under *x*-polarized incidence, since the charges could directly pass across the gap through the junction. Thus, it can be seen as a CTP mode. In this mode, the connected triangles serve as a single dipole which has a higher horizontal size, resulting in a stronger resonance strength at a lower frequency. However, under *y*-polarized incidence, the two triangles in both the DTD and DTDJ structures behave as two identical dipoles along the *y* direction. In this case, the charges are mainly driven to move along the *y*-direction, the junction no longer serve as a charge transfer link between the two triangles. Therefore, the corresponding transmission spectra are nearly the same.

The electric field distributions above also reveal a field enhancement effect in the DTD structures. The field enhancement is an important effect of metamaterials which could enhance the light-matter interaction in a subwavelength scale, and has potential applications in enhancing sensing and nonlinear effect. To investigate the field enhancement effect, we changed the gap distance of the DTD structures and monitored the electric field distributions at the BDP resonances. Figure 3a illustrates the simulated transmission spectra when the gap distance *g* decreases from 6 μm to 1 μm under *x*-polarized incidences, which show very little difference except for a minor red shift of the resonance frequency due to the decrease of the overall structure length. Figure 3b–g illustrates the corresponding simulated electric field distributions at the structure plane, it can be seen that the electric field at the gap becomes stronger as *g* decreases, which can be attributed to the enhanced capacitive coupling strength. The obtained maximum field enhancement factor *F*_e_ at *g* = 1μm reaches about 67.4, which is much higher than previous report [28]. Here, the field enhancement factor $F_{e}=\left|E_{\mathrm{sam}}\right|/\left|E_{ref}\right|$, where $\left|E_{\mathrm{sam}}\right|$ and $\left|E_{\mathrm{ref}}\right|$ are respectively the simulated electric field amplitude at the center point of the gap with and without the dimers. This strong field enhancement effect could be explained as follows. The *x*-polarized terahertz wave can excite nearly the same amount of charges with opposite sign at the ends of the gap, leading to a capacitance effect in the gap. When the distance between the two ends (the effective capacitor plates) decreases, the electric field in the gap would be greatly enhanced accordingly.

The large field enhancement effect is of great importance to design plasmonic sensors, modulators and switches [29,30,31]. Moreover, the field enhancement could induce larger nonlinear effect and meanwhile decrease the nonlinear modulation threshold, thus it is also a good strategy to design nonlinear devices. Here, we combine this field enhancement effect with graphene nonlinearity to show a dynamic resonance switching. As the conductivity of graphene can be easily controlled by external voltage, to leave the freedom of electric control, we use a graphene stripe to fill the gap of the DTD structure (DTDGS). Before that, we study the case of filling the gap with metallic stripe (DTDS). Figure 2e,f illustrate the corresponding measured and simulated transmission spectra, respectively. It can be seen that the transmission spectra of DTDS structures under *x*-polarized incidence are nearly the same as those of the DTDJ structures, which indicates the possibility in realizing electric control of the resonances when replacing the metallic strip with graphene. Under *y*-polarized incidences, big transmission change appears at the lower frequency range due to the large permittivity of metal. In this work, we only focus on nonlinear control for *x*-polarized incidence.

Here, we carried out numerical simulation to study the dynamic resonance switching effect using graphene stripe instead of metallic stripe (see the inset of Figure 2e). In the simulation, we fixed the gap distance *g* = 1 um to achieve a large field enhancement effect and also to make the carrier easy to transfer across the graphene. The graphene was modeled by conductivity (*σ* = *σ*_intra_ + *σ*_inter_) which can be described by the Kubo formula [32,33]: (1)$$\sigma_{\mathrm{intra}}=i\frac{e^{2}k_{B}T}{\pi \hbar^{2}(\omega +i\tau^{-1})}\left[\frac{E_{F}}{k_{B}T}+2\ln\left(\exp\left(-\frac{E_{F}}{k_{B}T}\right)+1\right)\right],$$
(2)$$\sigma_{\mathrm{inter}}=i\frac{e^{2}}{4\pi \hbar }\ln\left[\frac{2\left|E_{F}\right|-\hbar \left(\omega +i\tau^{-1}\right)}{2\left|E_{F}\right|+\hbar \left(\omega +i\tau^{-1}\right)}\right],$$
where *σ*_intra_ and *σ*_inter_ are respectively the conductivities contributed from the intraband electron-photon scattering and interband electron transition, *e* is the electron charge, *k_B_* is the Boltzmann constant, *T* is the temperature, ℏ is the reduced Planck’s constant, *τ* is the carrier scattering time, and *E_F_* is the Fermi level. In the terahertz range, the graphene conductivity is dominated by the intraband electron-photon scattering *σ*_intra_. According to former published article [10,34,35], the strong terahertz incident field mainly affect the carrier scattering time *τ* because of the field-induced redistribution of the electrons. Larger field strength corresponds to smaller carrier scattering time. Here, we fixed the Fermi level of graphene *E_F_* = 0.8 eV, which could be achieved by applying external bias [36,37]; meanwhile, varied the carrier scattering time *τ* from 80 to 10 fs, which are in the level of previously reported works [34,35]. The corresponding required incident terahertz fluence in practice can be roughly estimated by the saturable power transmission function (see the Appendix A). Here, parameters of *E_F_* = 0.8 eV and *τ* = 80 fs correspond to an incident terahertz fluence level of ~170 μJ/cm^2^, and parameters of *E_F_* = 0.8 eV and *τ* = 10 fs correspond to an incident terahertz fluence level of ~5 μJ/cm^2^. Such terahertz fluence range is easily to achieve using strong-field terahertz system in reference 36, whose maximum terahertz fluence reaches 190 μJ/cm^2^. Here, we run numerical simulations to obtain the response of the DTDGS structure under strong-field terahertz incidences. In the simulation, we assume the thickness of the single layer graphene is 1 nm and its permittivity *ε* = *ε*_r_ + i*ε*_i_ is calculated by *ε*_r_ = −*σ*i/(*ωdε*_0_) and *ε*_i_ = *σ*_r_/(*ωdε*_0_) [33,38], where *ω* is the angular frequency, *d* is the thickness of graphene, *ε*_0_ is the vacuum dielectric constant. Figure 4 illustrates the calculated permittivity of the graphene at different carrier scattering time. It is seen that as the carrier scattering time decreases, the real part of the permittivity increases while the imaginary part decreases, indicating a gradually decreasing conductivity. Figure 5a,b illustrate the corresponding simulated transmission and reflection spectra, respectively, by substituting the calculated graphene permittivity into the DTDGS structure with gap distance of 1 μm, as shown by the inset in Figure 5a. It can be seen that the CTP resonance mode gradually vanishes as *τ* decreases due to the fact that smaller *τ* corresponds to smaller graphene conductivity, which makes the charges harder to transfer across the gap of the two triangles. Thus, the strength of the CTP resonance mode is gradually decreases and even disappears. To reduce the effect of misalignment in real fabrication, we design the width of the graphene to be 10 μm, which is much larger than the gap distance of 1 μm. Further simulation results show that, as long as the misalignment is smaller than 3 μm (the fabrication error of conventional lithography is normally around 1 μm), the design could still function well. 

The above mentioned process was further confirmed by the simulated electric field distributions at the corresponding CTP resonances. As illustrated in Figure 6, when *τ* = 80 fs, corresponding to the lowest incident terahertz fluence, the electric field distribution is similar to that of the DTDJ structure, as illustrated by the left inset of Figure 2d, indicating the charges could pass through the gap due to the shoring effect by the large graphene conductivity. When *τ* decreases, corresponding to increasing the incident terahertz fluence, the electric field in the gap gradually increases, which means that the amount of charges that can pass through the gap reduces due to the decrease of the graphene conductivity, leading to a weaker CTP resonance.

Here, the nonlinear modulation depth of the proposed DTDGS structure at the CTP resonance is about 20%. To improve the nonlinear modulation depth, one route is to increase the range of the scattering time *τ*. However, this option mainly depends on several scattering mechanisms, including short-range neutral impurity scattering, charged long-range impurity scattering, scattering due to absorption of optical phonons, and the scattering rate due to interactions with acoustic phonons and carrier-carrier scattering, while even the substrate could also have an effect on the carrier scattering time [35,39]. Therefore, it is quite difficult to control *τ*. Another route to increase the nonlinear modulation depth using the current settings involves applying the graphene to modulate structures to support high-*Q* resonances, such as dielectric structures, since their resonance strengths are quite sensitive to the changes of the external environment.

## 4. Conclusions

In conclusion, we experimentally and theoretically investigated the mode switching effect from BTP resonance mode to CTP resonance mode in triangular dimer metamaterials with or without a junction in the terahertz regime. We also study the field enhancement effect of the BTP resonance at the gap in the DTD structure. Based on these investigations, we proposed a dynamic mode switching design by integrating graphene stripe into the gap and applying the nonlinear effect of graphene. We found that the CTP plasmonic resonance could be effectively tuned by the incident terahertz field strength. The proposed method may find applications in terahertz sensing and nonlinear amplitude modulating.

## Figures and Tables

**Figure 1 materials-12-02466-f001:**
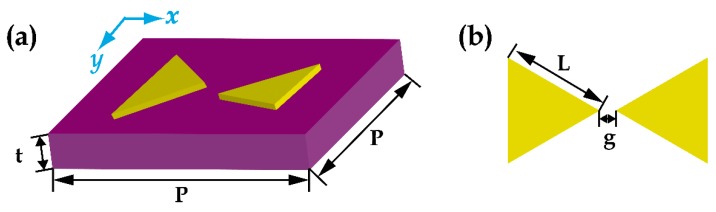
(**a**,**b**) Schematics of the DTD structure.

**Figure 2 materials-12-02466-f002:**
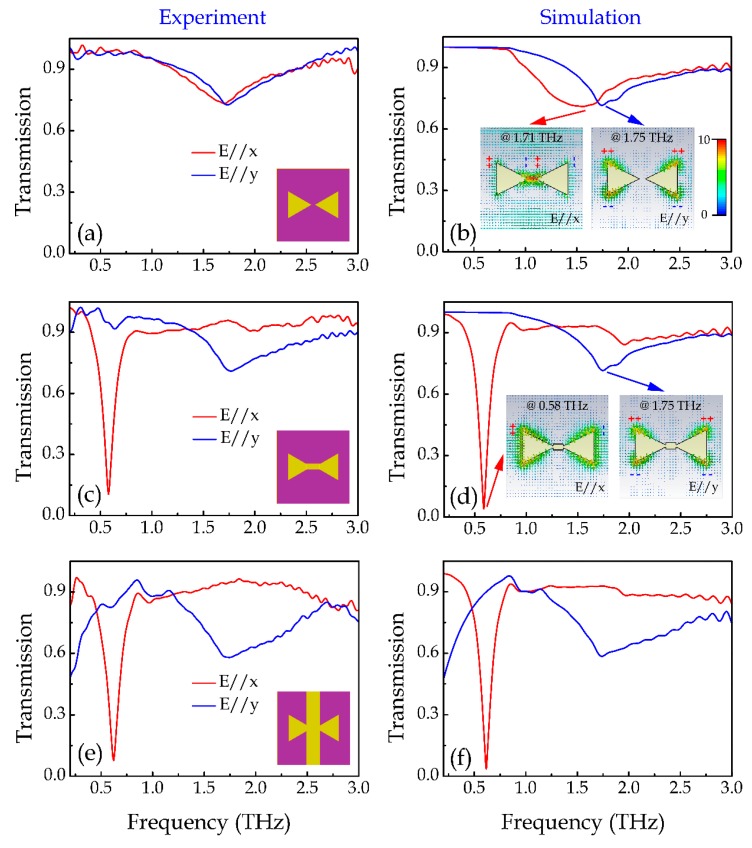
(**a**,**c**,**e**) Measured transmission spectra of the DTD, DTDJ and DTDS structures under *x*-polarized (red) and *y*-polarized (blue) incidences, respectively. The insets are the schematics of the corresponding structures. (**b**,**d**,**f**) Simulated transmission spectra of the DTD, DTDJ and DTDS structures under *x*-polarized (red) and *y*-polarized (blue) incidences, respectively. The insets in (**b**) and (**d**) are the simulated electric field distributions at the corresponding resonances, as indicated by the inset arrows. The “+” and “−” in the insets represents the sign of the accumulated charges at the ends of the corresponding structures. All the inset field distributions in (**b**,**d**) share a same color bar in (**b**).

**Figure 3 materials-12-02466-f003:**
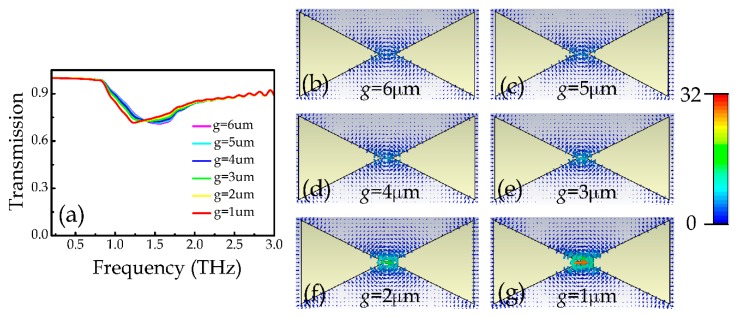
(**a**) Simulated transmission spectra of the DTD structures with different *g* from 6 to 1 μm under x-polarized incidences, respectively. (**b**–**g**) Simulated electric field distributions of the DTD structures with different *g* from 6 to 1 µm at the corresponding BDP resonance frequencies under *x*-polarized incidences, respectively.

**Figure 4 materials-12-02466-f004:**
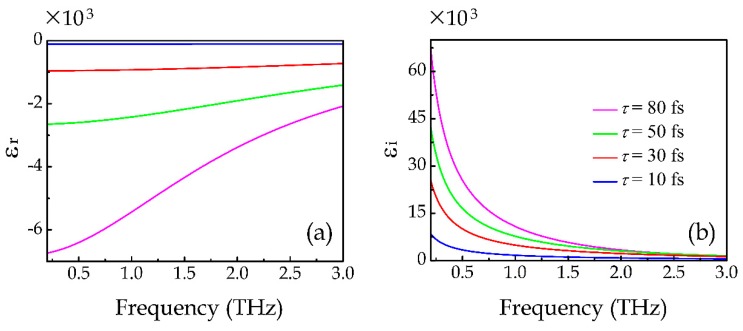
Calculated (**a**) real parts and (**b**) imaginary parts of graphene permittivity.

**Figure 5 materials-12-02466-f005:**
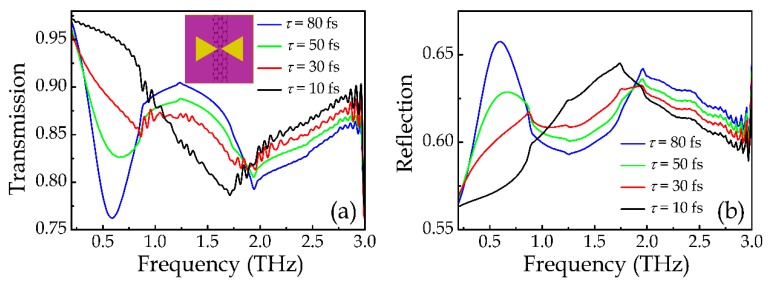
(**a**) Simulated transmission spectra of the DTDGS structure with *g* = 1 μm under *x*-polarized incidences at different graphene scattering time *τ* from 80 to 10 fs, respectively. (**b**) Corresponding simulated reflection spectra of the DTDGS structure. The insets in (**a**) is the schematic of the corresponding structure.

**Figure 6 materials-12-02466-f006:**
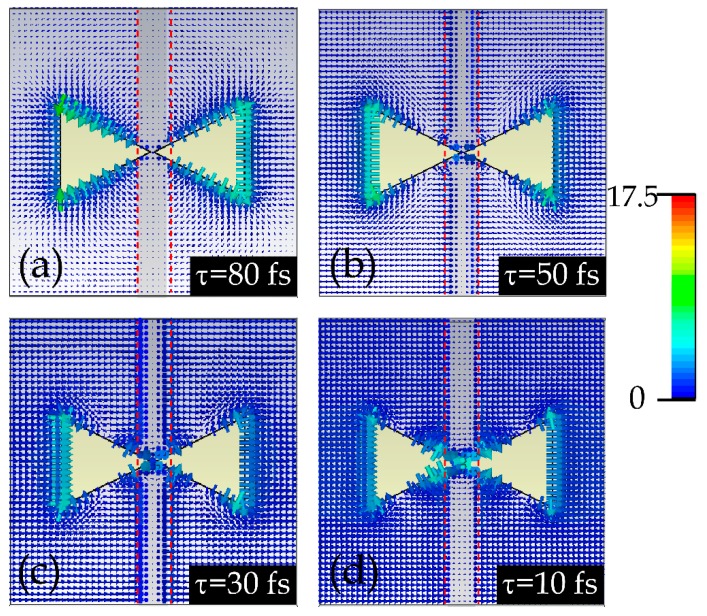
(**a**–**d**) Simulated electric field distributions of the DTDGS structures with *g* = 1 um under *x*-polarized incidences at different graphene scattering time *τ* from 80 to 10 fs, respectively. The red dotted lines represent the boundaries of the graphene stripes.

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
