# Peer review of "Nonlinear Modulation of Plasmonic Resonances in Graphene-Integrated Triangular Dimers at Terahertz Frequencies"

_materials, 2019, doi:10.3390/ma12152466_

Round 1
Reviewer 1 Report
The revised paper answered all the issues raised by the reviewer carefully. In particular, the detailed comments on the scattering time become very useful for the readers. I recommend this paper can be published as it is.
Reviewer 2 Report
The paper is intersting, presents sufficient modeling results. It would be very good to check the predicted results experimentally then.
This manuscript is a resubmission of an earlier submission. The following is a list of the peer review reports and author responses from that submission.
Round 1
Reviewer 1 Report
In this work, authors have reported about the active switching of plasmonics modes in triangular dimers using graphene. Although the mode switching using metal junction is already reported, the motivation being active switching of these modes is promising. However, there is no experimental data that proves the active switching, which is the key point of this paper. The physical phenomena behind the active switching should be discussed more. In this case, authors have focused more on the metal junction dimers. Moreover, the authors have lots of mistakes while referring figures in the manuscript. Hence I recommend this paper to be rejected.
Reviewer 2 Report
This paper propose an active switching device based on two plasmonic modes excited in the triangular dimer structure by adjusting the conductivity of graphene. By changing scattering time of the graphene, forming metallic junction enables the plasmonic mode, BTP, to change into the CTP mode, which is interesting idea to control plasmonic modes actively. However, the values of the key parameter, scattering time, should be revised because there is no clear explanation for the values.
In line 166-168, the scattering time changes from 0.1 to 0.4 ps by changing the intensity of the incident terahertz field. (1) Authors should write down the exact relation or equations between the scattering time and the intensity of the THz field. In addition, proper references for such equations should be cited. In the paper, there is no background information how such scattering time is assumed. (2) Authors should comment on the feasibility of the estimated intensity of the incident THz field to achieve the assumed scattering time.
In the gap, if the bias of the graphene is controlled, what effects will be observed?
In the real situation, the misalignment between the graphene stripe and the gap of the DTD structure can be easily observed. If such fabrication error is assumed, authors can comment on the performance of the device.
In line 128, according to the authors, the field enhancement of 67.4 is much higher that previous paper. Can authors explain why higher enhancement is observed in their structure?
Reviewer 3 Report
The paper “Switching Terahertz Plasmonic Resonance Modes in 3 Triangular Dimers using Graphene” presents an idea to approach to tunable THz – IR –optical structure using tunability of graphene. The paper presents some interesting results and predictions, however could not be published before major revision. It is necessary to give much more details concerning the experiment and simulations:
1. It is not clear what experimental samples are: substrates, graphene quality and origin, graphene doping level, grain size, graphene boudaries, etc.
2. The THz experimental errors in transmission mode.
3. the level of reflection in THz range, if possible (either measurmeent or modeling).
4. The simulation details including the used method for numerical simulation.
5. How do you plan control the graphene scattering time in practice?